# A dataset of lake-catchment characteristics for the Tibetan Plateau

Junzhi Liu[1,2*], Pengcheng Fang[2,3], Yefeng Que[2,3], Liang-Jun Zhu[4], Zheng Duan[5], Guoan Tang[2,3], Pengfei Liu[1], Mukan Ji[1], Yongqin Liu[1,6]

[1]Center for the Pan-Third Pole Environment, Lanzhou University, Lanzhou, 730000, China
[2]Jiangsu Center for Collaborative Innovation in Geographical Information Resource Development and Application, Nanjing, China
[3]Key Laboratory of Virtual Geographic Environment (Nanjing Normal University), Ministry of Education, Nanjing, 210023, China
[4]State Key Lab of Resources and Environmental Information System, Institute of Geographic Sciences and Natural Resources Research, CAS, Beijing, 100101, China
[5]Department of Physical Geography and Ecosystem Science, Lund University, Lund, 22100, Sweden
[6]State Key Laboratory of Tibetan Plateau Earth System, Resources and Environment, Institute of Tibetan Plateau Research, Chinese Academy of Sciences, Beijing, 100101, China

*Correspondence to*: Junzhi Liu (liujunzhi@lzu.edu.cn)

**Abstract.** The management and conservation of lakes should be conducted in the context of catchment because lakes collect water and materials from their upstream catchments. So the datasets of catchment-level characteristics are essential for limnology studies. Lakes are widely spread on the Tibetan Plateau (TP), with a total lake area exceeding 50 000 km$^2$, accounting for more than half of the total lake area in China. However, there has been no dataset of lake-catchment characteristics in this region to date. This study constructed the first dataset of lake-catchment characteristics for 1525 lakes with areas from 0.2 to 4503 km$^2$ on the TP. Considering that large lakes block the transport of materials from upstream to downstream, lake catchments are delineated in two ways: the full catchment, which refers to the full upstream contributing area of each lake, and the inter-lake catchments, which are obtained by excluding the contributing areas of upstream lakes larger than 0.2 km$^2$ from the full catchment. There are six categories (i.e., lake body, topographic, climatic, land cover/use, soil & geology, and anthropogenic activity) and a total of 721 attributes in the dataset. Besides multi-year average attributes, the time series of 16 hydrological and meteorological variables are extracted, which can be used to drive or validate lumped hydrological models and machine learning models for hydrological simulation. The LCC-TP dataset contains fundamental information for analysing the impact of catchment-level characteristics on lake properties, which on the one hand, can deepen our understanding of the drivers of lake environment change, and on the other hand can be used to predict the water and sediment properties in unsampled lakes based on limited samples. This provides exciting opportunities for lake studies in a spatially-explicit context and promotes the development of landscape limnology on the TP. The dataset of lake-catchment characteristics for the Tibetan Plateau (LCC-TP v1.0) is accessible at the National Tibetan Plateau/Third Pole Environment Data Center (https://doi.org/10.11888/Terre.tpdc.272026, Liu, 2022).

# 1 Introduction

Lakes are an essential component of inland water and play a key role in maintaining regional ecosystem services (Cole et al., 2007). The management and conservation of lakes should be conducted in the context of catchment because lakes collect water and materials from their upstream catchments. The properties of lake water and sediments (e.g., nutrient concentrations and carbon storage) are affected by catchment-level characteristics such as terrain, land cover, and precipitation amount (Soranno et al., 2010). It was reported that catchment-level land-use composition could explain 45-62%

of lake water-quality metrics (e.g., turbidity, total nitrogen, and dissolved organic carbon) across conterminous US (CONUS) (Read et al., 2015). Therefore, characterizing the upstream catchments of lakes is essential for the scientific study and management of lakes.

It requires multiple steps and specialized geospatial techniques to calculate catchment-level characteristics (Hill et al., 2018; Hao et al., 2021). First, flow directions should be determined from digital elevation model (DEM), and catchment boundaries

are then delineated according to flow direction. After that, multiple related spatial datasets are collected and processed (e.g., data format conversion and reprojection). Finally, zonal statistics are performed to get catchment-level characteristics. These procedures need to be repeated for every lake in a region, which is time-consuming, so automatic processing needs to be implemented. This is not easy for people who are not experts on geospatial techniques. In addition, the lake-catchment characteristics calculated by different researchers are usually not consistent on the aspects of feature types and data sources,

making the analysis based on them less comparable.

To provide consistent baseline datasets of lake-catchment characteristics, several products, such as the LAGOS-NE and Lake-Catchment (LakeCat) datasets (Soranno et al., 2017; Hill et al., 2018), have been produced. The LAGOS-NE dataset contains catchment-level characteristics for 51 101 lakes and reservoirs larger than 4 ha in the northeastern-most 17 US states. In this dataset, lake catchments were defined as "inter-lake watersheds" which contains two parts: the area draining

directly into a lake and the area draining into its upstream streams and lakes smaller than 0.1 km$^2$ (Soranno et al., 2017). The contributing areas of upstream lakes larger than 0.1 km$^2$ were not included because large lakes can block the transport of materials from upstream to downstream (Zhang et al., 2012). The LakeCat dataset, as an extension to LAGOS-NE, covers the CONUS and contains the data for 378 088 lakes. Besides inter-lake watersheds, the whole upstream watershed was also used as the statistical units and there were more than 200 catchment-level attributes characterizing soil, lithology, land cover,

mines, roads, and so on (Hill et al., 2018). These datasets facilitated the research on landscape limnology, which means the study of lakes in the context of catchment-level landscapes (Soranno et al., 2010). Besides, the river-oriented datasets of catchment characteristics, represented by the CAMELS (Catchment Attributes and Meteorology for Large-sample Studies) series of datasets like CMALES (Addor et al., 2017), CAMLES-CL (Alvarez-Garreton et al., 2018), CMALES-BR (Chagas et al., 2020), CAMLES-GB (Coxon et al., 2020), CCAM (Hao et al., 2021), and LamaH-CE (Klingler et al., 2021), also

showed the great value of such catchment-level attributes datasets.

In this research, we focus on the Tibetan Plateau (TP), which has a total lake area exceeding 50 000 km$^2$, accounting for more than half of the total area of lakes in China (Zhang et al., 2019). Due to the paucity of in-situ measurements in lakes on the TP, catchment-level characteristics are especially important as they can be used to predict water and sediment properties in unsampled lakes based on limited samples. However, there has been no dataset of lake-catchment characteristics on the

TP yet, which hinders the research on lakes in this region. This study aims to construct the first dataset of lake-catchment characteristics for the Tibetan Plateau (LCC-TP v1.0) and provide indispensable data for studies on TP lakes. Section 2 introduces the study area. Section 3 describes the methodology for metrics calculation of lake-catchment characteristics, including catchment delineation, attribute data collection and zonal statistics. The main lake-catchment characteristics on the TP are presented in Section 4. Section 5 concludes and discusses the potential application of the constructed dataset.

**2 Study area**

The TP is located between 74–98° E and 28–40° N (Fig. 1). It is the largest and highest plateau in the world, covering an area of about 2.5×10$^6$ km$^2$ and with an average altitude above 4000 meters above sea level (Zhang et al., 2019). The TP is the source of more than 10 big rivers, such as the Yangtze River, the Yellow River, and the Ganges River, therefore also acknowledged as the "Water Tower of Asia" (Immerzeel et al., 2010; Gao et al., 2021). Figure 1 shows the major basins

over the TP, such as Brahmaputra, Hexi Corridor, Indus, Inner TP, Mekong, Qaidam, Salween, Tarim, Yangtze River, and Yellow River. Lakes are a key component of the Asia water tower, and there are 1 424 lakes with an area of more than 1 km$^2$ (Zhang et al., 2019). Most lakes on the TP are seldom disturbed by human activities, so they are good information carriers of global changes in this region (Li et al., 1998).

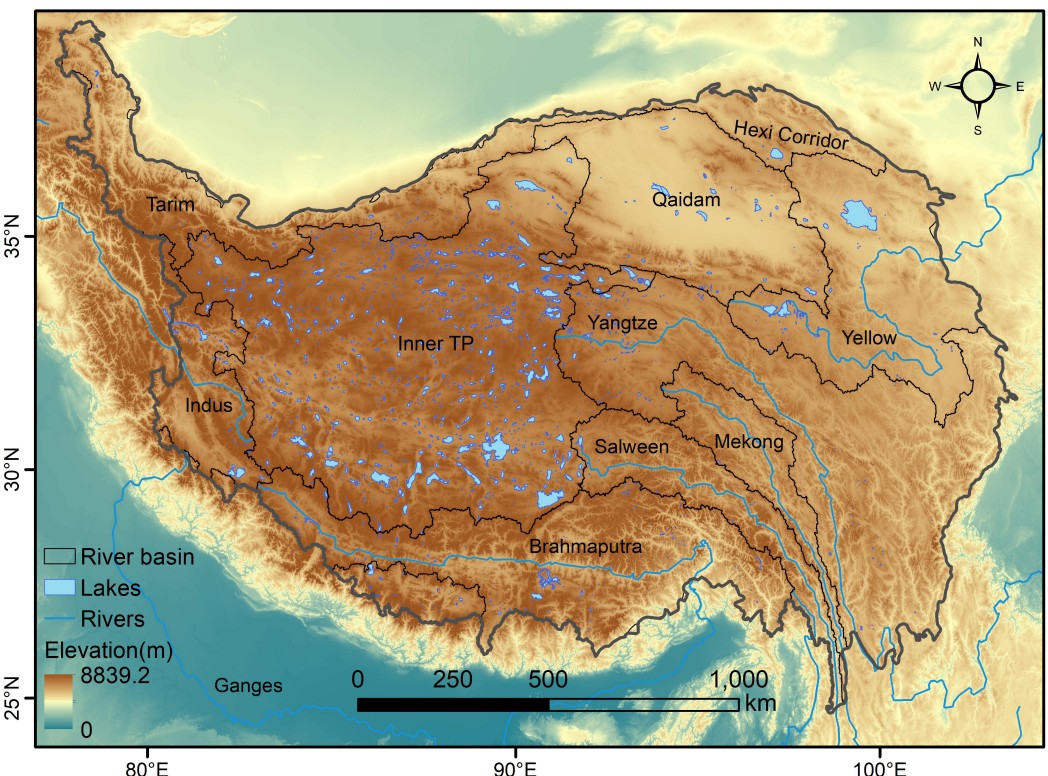

**Figure 1: The spatial distribution of lakes on the Tibetan Plateau.**

## 3 Methodology and source datasets

Three steps are conducted to construct the LCC-TP dataset (Fig. 2). First, delineate the lake catchments and establish the topological relationships among nested lakes. Meanwhile, collect related attribute datasets and conduct necessary processing such as data format conversion and reprojection. Then perform zonal statistics using catchment extents and datasets of spatial attributes to obtain the catchment-level attributes.

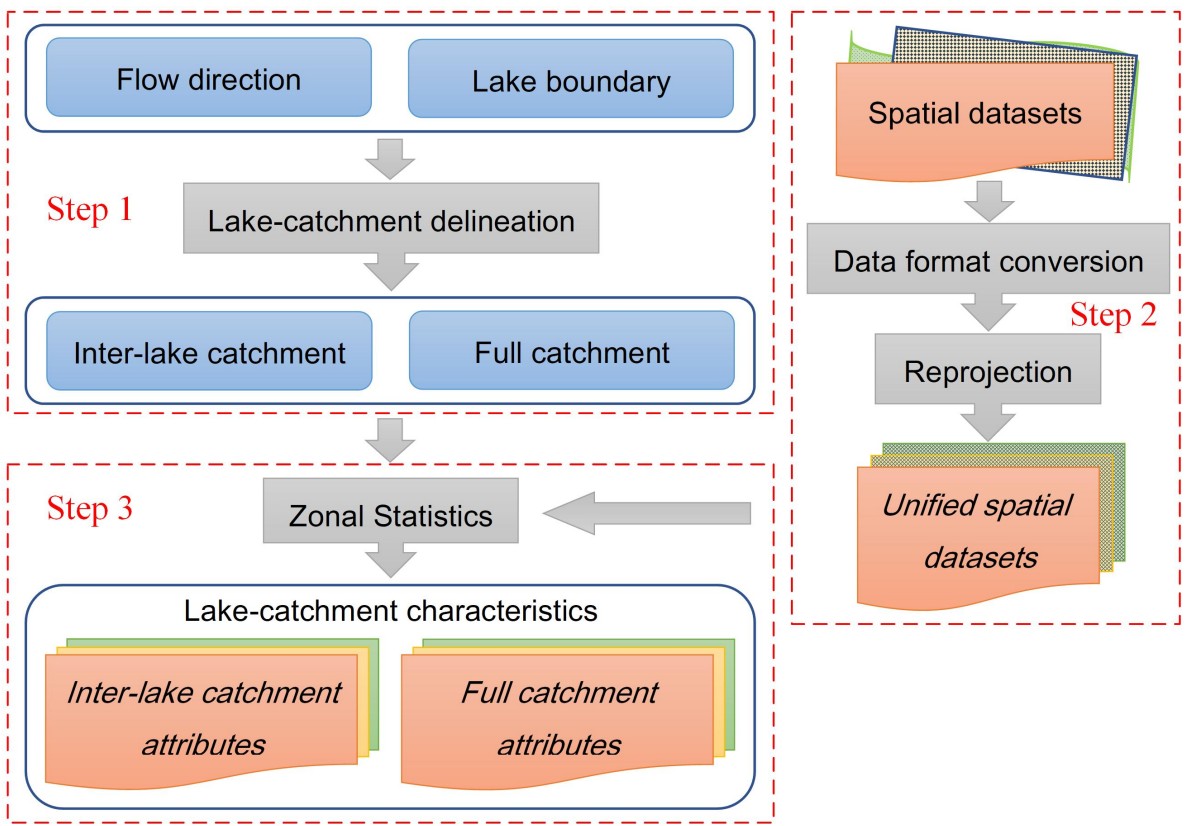

**Figure 2: Procedures to construct the dataset of lake-catchment characteristics for the TP (LCC-TP).**

## 3.1 Catchment delineation

Considering that large lakes are likely to block the transport of materials from upstream to downstream, two types of catchments, the full catchments and inter-lake catchments, are defined in this study. The full catchment refers to the full upstream contributing area of each lake, while the inter-lake catchment is obtained by excluding the contributing areas of upstream lakes larger than 0.2 km$^2$ from the full catchment following the definition of Soranno et al. (2017). For example, the green area in Fig. 3 is the full watershed of lake No. 1, and the black stippled area is its inter-lake catchment.


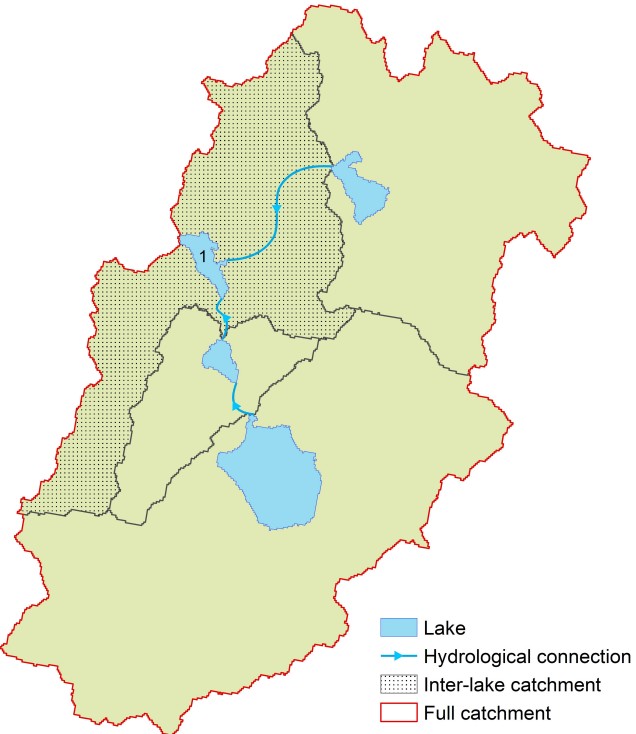

Lake
Hydrological connection
Inter-lake catchment
Full catchment

**Figure 3: Illustration of the inter-lake catchment (black stippled area) and full catchment (green area).**

Traditional river-oriented catchment delineation methods are not suitable for the delineation of lake catchments. Liu et al (2020) proposed a lake-oriented approach to delineating endorheic catchments, which can be used to delineate the full catchments of endorheic lakes in this study. But there are more tasks in this study, including the delineation of both full catchments and inter-lake catchments for endorheic lakes and upstream lakes, the construction of topological relationship among lakes/lake-catchments, and the tracing of flow path among upstream and stream lakes. Therefore, we developed a software using the C and Python programming language to implement these tasks, and the source code are open (https://github.com/LoserOne-ovo/basin_delineation).

Flow direction and lake boundary data are needed for catchment delineation. Firstly, the vector lake data is rasterized using the same geospatial reference system and pixel size as the flow direction data (Yamazaki et al., 2019). Meanwhile, calculate the reverse flow direction data (recorded as 8 bits corresponding to 8 neighbours, e.g., 10000001 means the first and eighth neighbours flow into the current pixel) to assist the tracing of upstream contributing areas. Next, iterate over all the pixels to find the inlets to the lake, which are defined as the pixels flowing directly into a lake according to the flow direction. Then put the inlet pixels into a stack. Repeat taking a pixel out and pushing its direct upstream pixels in until the stack is empty. Finally, we get the full upstream contributing area (i.e. full catchment) of the lake. Following the same procedures above but

adding a termination criterion that if a tracing branch meets an upstream lake pixel then return, the inter-lake catchments can be obtained.

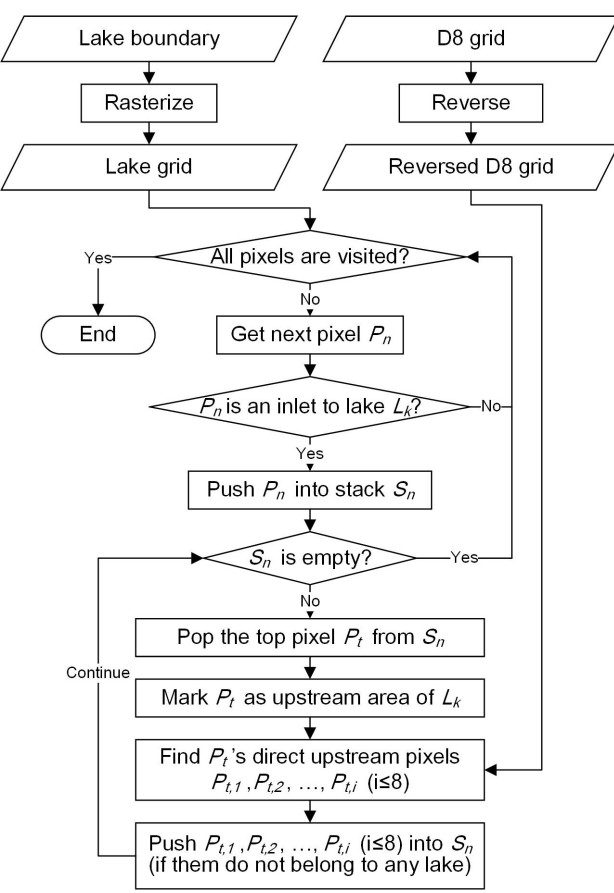

**Figure 4: Flowchart for lake catchment delineation.**

In this study, flow direction data from the MERIT Hydro dataset (Yamazaki et al., 2019) was used, which has a resolution of
3 arc-second (~90 m at the equator). This dataset was verified to have high accuracy in terms of flow accumulation area and river basin shape. The lake boundaries in 2018 were used in the delineation of catchments, and the dataset from Zhang et al. (2019) (https://doi.org/10.11888/Hydro.tpdc.270303) was adopted. Some additional operations like merging adjacent lakes and filling holes were employed to remove topology errors.

### 3.2 Data collection and processing

The lake-catchments are characterized from six aspects: lake body, topography, climate, land cover/use, soils & geology, and anthropogenic activities. The following standards were adopted for source dataset selection. Firstly, the corresponding environmental variables should have theoretical impacts on the attributes of lake water or sediment. Secondly, the dataset covers the whole TP and can be publicly available. Finally, the most reputable datasets with high resolution are preferred

when multiple data sources are available. Following these standards, the vast majority of selected datasets were well-validated and recognized in each field. For example, the SoilGrids dataset (Poggio et al., 2021) is the de facto standard of gridded soil property data, and the China meteorological forcing dataset (CMFD) (Yang and He, 2019) is the most widely used climatic dataset on the TP. The information of source datasets used in the LCC-TP dataset can be found in Table A1. All the spatial datasets were converted to GDAL/OGR readable formats and projected to the Albers equal-area conic projection. Besides static attributes, the dynamic time series of hydrological and meteorological data are also provided. Version 1.0 of LCC-TP offers 57 different variables with a total of 721 individual attributes. The details of these attributes are described in Section 4.

## 3.3 Zonal statistics

For each lake, zonal statistics are conducted to calculate catchment-level characteristics for inter-lake catchments and full catchments, respectively. Different methods are used for raster data and vector data. For raster data, the grid cells within each catchment are picked out and the statistics of their values are calculated. For continuous variables (e.g. elevation and precipitation amount), the average value or maximum/minimum value is calculated. For categorical variables (e.g. land use type), the mode is calculated, and the percentages of each category are also calculated for some variables. The *rasterstats* Python package (https://github.com/perrygeo/python-rasterstats) and the Google Earth Engine cloud platform were adopted for implementation. For vector data like the spatial distribution of glaciers stored in the ESRI shapefile format, it is first intersected with the catchment extent layer, then the ratio of the intersection's area within each catchment to the catchment's total area is calculated. The obtained spatial distribution of catchment-scale variables was plotted and visually checked by the authors to ensure the correctness of the dataset. For numerical variables whose original data was in raster format, their ranges at the grid-cell scale (i.e. before zonal statistics) and the catchment scale (i.e. after zonal statistics) were calculated (Table S2), and it is checked that the range at the catchment scale fell within those at the grid-cell scale for each variable.

## 4 Results

### 4.1 Validation of the delineated catchments

The catchment delineation in this study was based on the flow direction data from Yamazaki et al. (2019) which has been widely verified. To further validate the accuracy of delineated catchments, the dataset from Liu et al. (2020) was used as a reference, which contains the boundaries of 421 lake catchments on the Inner TP. Figure 5 shows that the catchment areas in this study have a high correlation (r=0.988) with those of Liu et al. (2020), which proves the correctness of our results. The small differences between these two datasets may be related to the errors in DEM and the different methods for depression filling and flow direction correction.

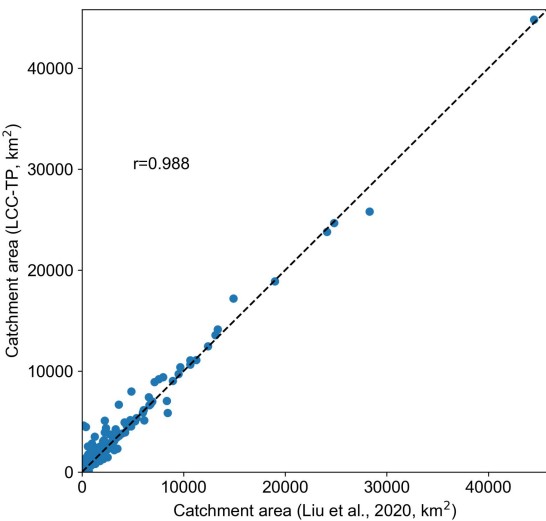

**Figure 5: Comparison of the areas of delineated catchments in this study with those of Liu et al. (2020).**

 **4.2 Lake body characteristics**

The area, perimeter, lake development index, and type of each lake were calculated. The lake development index was used to characterise the complexity of lake shoreline, which was defined in formula (1), where $L$ represents the length of lake shoreline and $s$ represents lake area, i.e. the ratio of the shoreline length to the circumference of a circle with the same area as the lake. The value increases with increasing shoreline complexity, and the maximum value is one while the shape is a circle.

$$dev = \frac{L}{2\sqrt{\pi s}} \#(1)$$

The type of lake herein refers to whether it is an upstream lake (i.e. a lake with outflow to a downstream river or lake) or a terminal lake (i.e. a lake without outflows).

The smallest lake has an area of 0.2 km$^2$ and the largest lake (i.e. Qinghai Lake) has an area of 4503.5 km$^2$. 72% of the 1525 lakes have an area less than 10 km$^2$. Figure 6 shows the spatial distribution of lake development index and type on the TP. The average development index for lakes across the TP is 3.40, and there is a cluster of lakes with high development indices 170 in the north of the Inner TP. 364 out of 1525 lakes (24 %) are terminal lakes, most of which are located in the Inner TP and the Qaidam Basin.

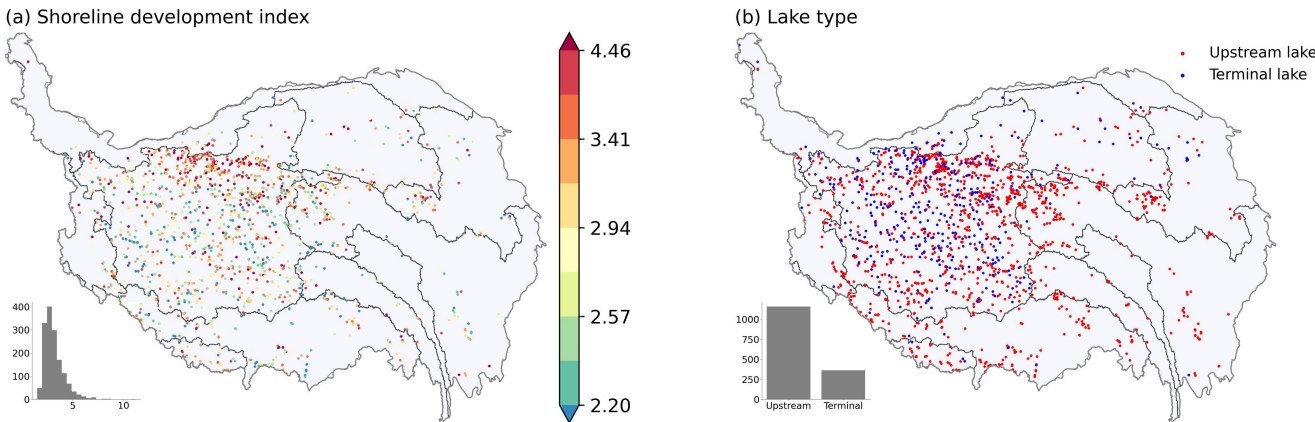

**Figure 6: Spatial distribution of lake development index (a) and type (b) on the TP.**

## 4.3 Topographic characteristics

The catchment-level elevation (including average, the maximum and minimum value), relief, slope, catchment area, and lake-catchment area ratio were calculated based on the MERIT DEM (Yamazaki et al., 2017). These characteristics were calculated for both the full and inter-lake catchments. The slope values were calculated using ArcGIS 10.5. The relief values, defined as the difference between the maximum and minimum elevations in a neighbourhood, were calculated using window sizes of $5 \times 5$, $11 \times 11$, $21 \times 21$, $31 \times 31$, $41 \times 41$, and $51 \times 51$ based on DEM (digital elevation model) of $0.00833°$

resolution. The usage of different window sizes in relief value calculation aims to meet the needs of different analysis scenarios: for the researches focusing on small-scale terrain variation, a small window size is appropriate; when the focus is large-scale terrain variation, a larger window size is preferred. Figure 7 shows the spatial distribution of mean elevation, relief, slope (%), and lake-catchment area ratio for inter-lake catchments on the TP. The mean elevation is relatively low in the eastern TP and the valley between the Kunlun and Gangdise Mountains in the south of the Inner TP. The relief and slope

are relatively low in the north of the Inner TP, where the elevation is very high. The lake-catchment area ratio is high in the south and east parts of the Inner TP and the upper Yellow River Basin.

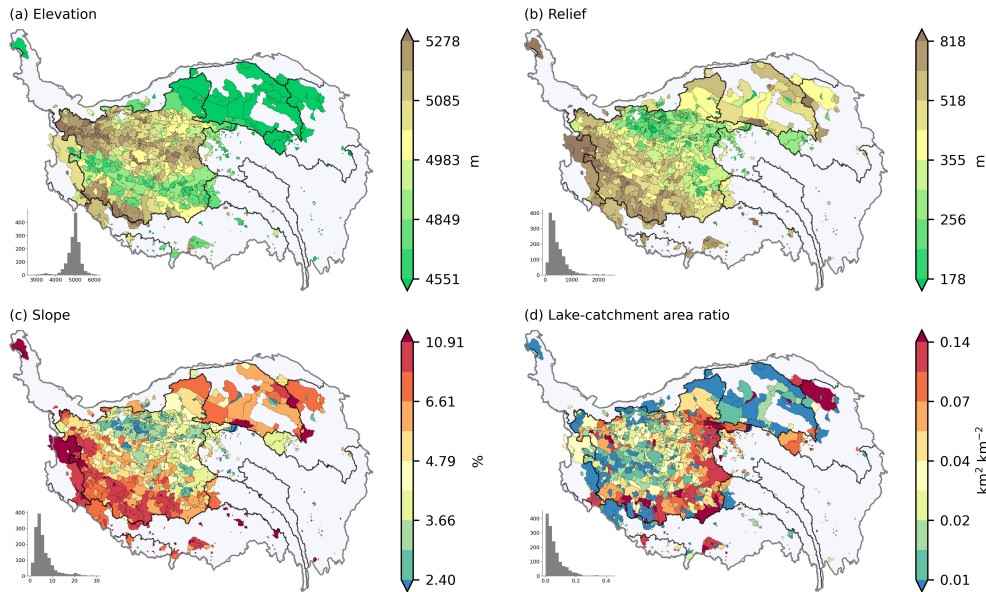

**Figure 7: Spatial distribution of mean elevation (a), relief (b), slope (%) (c), and lake-catchment area ratio (d) for inter-lake catchments on the TP. The relief was calculated using window size of 11×11 based on DEM of 0.00833° resolution.**

### 4.4 Climatic characteristics

Eleven climatic variables were included in the constructed dataset, including 2-meter air temperature, surface pressure, and specific humidity, 10-meter wind speed, downward shortwave radiation, downward longwave radiation, precipitation amount, potential evapotranspiration (PET), actual evapotranspiration (AET), climate moisture index (CMI), and aridity index. The multi-year average values of all the variables were calculated at three levels (i.e. the lake body, inter-lake catchment, and full catchment level), and the monthly and growing-season (May to September) average values of all the variables except the aridity index were also calculated.

The grid-based CMFD dataset (Yang and He, 2019), ranging from 1979 to 2018, was used to calculate the catchment-level climatic characteristics. CMFD was constructed through the fusion of in-situ observations from weather stations, remote sensing products, and reanalysis datasets, which improved the data quality in Western China where weather stations are sparse. It has a spatial resolution of 0.1° and a temporal resolution of three hours. The background field data of air temperature came from GLDAS NOAH10SUBP 3H V001 and the precipitation data was the combination of GLDAS NOAH10SUBP 3H V001, GLDAS NOAH025 3H V2.1 and TRMM 3B42 V7.

PET was derived from the Global Potential Evapotranspiration (Global-PET) dataset (Zomer et al., 2008). In this dataset, monthly PET was estimated via the Hargreaves (1994) equation at a spatial resolution of 30 arc seconds using precipitation and temperature inputs obtained from the WorldClim dataset (Hijmans et al., 2005). The aridity index was derived from the Global-Aridity dataset (Zomer et al., 2008), which quantifies precipitation availability over atmospheric water demand and was calculated as the ratio of long-term mean precipitation and PET. CMI was another metric to characterize the degree of

humidity, which is defined via following function: [CMI = (P / PET) - 1 when P < PET] or [CMI = 1 - (PET / P) when P >= PET] (Willmott and Feddema, 1992).

Figure 8 shows the spatial distribution of multi-year average climatic characteristics for inter-lake catchments on the TP. The air temperature and pressure are low in the north of Inner TP where elevation is high. Radiation is high in the southwest of the TP and low in the north part. Wind speed is high in the east and southwest of the Inner TP. Precipitation and evapotranspiration have a decreasing trend from southeast to northwest, and accordingly, it gets drier from southeast to the northwest as shown by the spatial distribution of air specific humidity, climate moisture index, and aridity index. It should be

noted that aridity indices are higher under more humid conditions and lower under more arid conditions according to its formula.

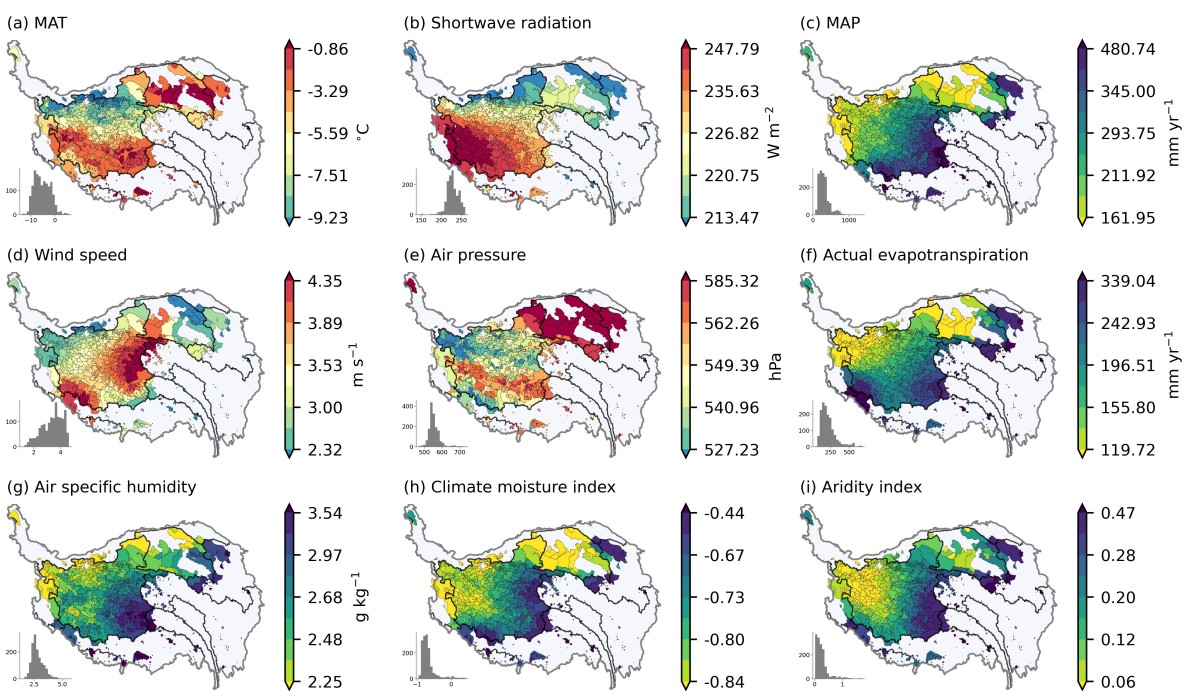

**Figure 8: Spatial distribution of multi-year average climatic characteristics for inter-lake catchments on the TP. MAT represents mean annual temperature, and MAP represents mean annual precipitation.**

**4.5 Land cover/use characteristics**

The land cover/use characteristics include remote-sensed vegetation indices (i.e. EVI (Enhanced Vegetation Index) and NDVI (Normalized Difference Vegetation Index)), Gross Primary Productivity (GPP), Net Primary Production (NPP) and dominant land cover/use type in each catchment and the fractions of each type, and the fractions of protected area. The average vegetation indices, GPP and NPP across the whole year and in the growing season were calculated, respectively.

The land cover/use data came from the fusion land use product on the TP produced by Xu (2019), which was constructed based on six mainstream land use products, i.e. ESA GlobCover (Arino et al., 2010), NLCD-China (Liu et al., 2005),

FROM-GLC (Gong et al., 2019), MODIS MCD12Q1 (Friedl and Sulla-Menashe, 2019), UMD GLCF GLCDS (Hansen et al., 1998), and USGS IGBP DISCover (Loveland et al., 2009). This dataset has a spatial resolution of 300 m and covers three historical periods (1992, 2005, and 2015). There are nine different land use types over the TP, including grassland, shrubland, forest, glacier, bare land, water body & wetland, desert, farmland and urban land. In addition, the second glacier inventory dataset of China (version 1.0, 2006-2011) (Liu et al., 2012) and a wetland distribution dataset (the 1970s, 2000s) (Zhou, 2018) were used as independent datasets for glaciers and wetlands. For the protected area, the World Database on Protected Areas (WDPA) (UNEP-WCMC and IUCN, 2021) was used. The fractional snow cover data (i.e. the fraction of a pixel that is snow-covered) was extracted from the MODIS daily cloud-free snow cover product over the Tibetan Plateau (2002-2015) (Qiu, 2018a). Considering that cloud and snow have similar reflection signals, eight different methods were employed in this product to remove the influence of cloud on snow cover identification.

Figure 9 shows the spatial distribution of land use/cover characteristics for inter-lake catchments on the TP. EVI is generally low for catchments across the TP and there is a deceasing trend from southeast to northwest following the spatial pattern of precipitation. The fraction of grassland is higher in the south of the Inner TP and the source region of the Yellow River, and that of shrubland is higher in the middle of the Inner TP. Wetlands have higher coverage mainly in the south and east part of the Inner TP and the upper Yellow River Basin. There is no cropland in most lake-catchments, and the bare land and desert are mainly distributed in the north of the TP. The faction of glaciers is relatively higher in the south and west of the TP.

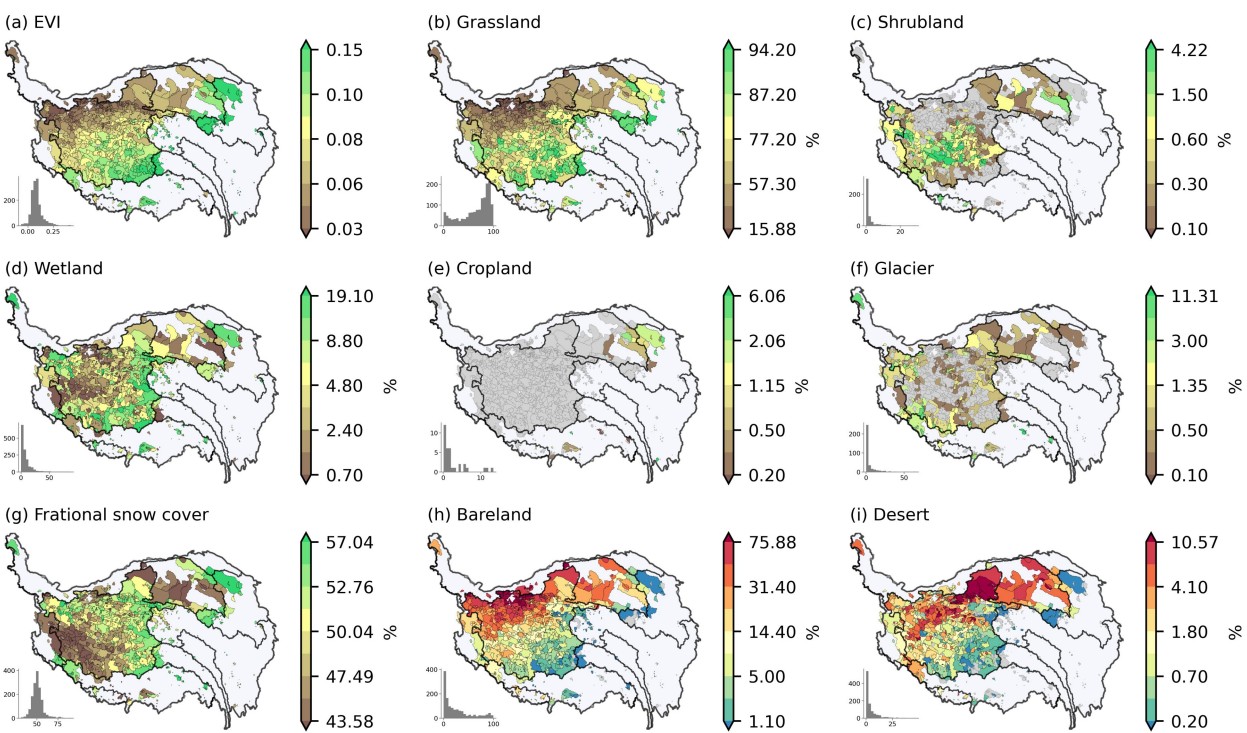

**Figure 9: Spatial distribution of land use/cover characteristics for inter-lake catchments on the TP. The multi-year average EVI from 2000 to 2021, the land use/cover type and fraction in 2015 are shown in the figure.**

## 4.6 Soil & geology characteristics

Twenty-one physical and chemical variables of soil were included in this study. The proportions of sand, silt, clay, and coarse fragments, the bulk density, cation exchange capacity (CEC), pH, total nitrogen (TN), and SOC (soil organic carbon) content/density/stock were derived from the 250 m-resolution SoilGrids 2.0 product (Poggio et al., 2021). These soil properties were predicted at six different depths (i.e. 0–5 cm, 5–15 cm, 15–30 cm, 30–60 cm, 60–100 cm, and 100–200 cm) using machine learning models based on observations from over 230 000 soil profiles globally in the WoSIS database and over 400 environmental covariates. Data of all the six layers were aggregated at the inter-lake catchment and full catchment level. The grading standards of soil particle size are as follows: coarse fragments in the soil refer to particles larger than 2 mm and smaller than 25 cm in diameter, sand refers to particles > 0.05 mm in the fine earth fraction (i.e. the particles less than 2 mm), silt refers to particles $\geqslant$ 0.002 mm and $\leqslant$ 0.05 mm in the fine earth fraction, and clay refers to the particles < 0.002 mm. The CEC measures the ability of soil to hold on exchangeable cations which can resist nutrient leaching and is an important variable in soil fertility. The organic carbon content, density, and stock are the mass of organic carbon per unit mass, volume, and surface area, respectively. Besides SoilGrids, the dataset of soil organic carbon content at different depth intervals (i.e. 0–30 cm, 0–50 cm, 0–100 cm, 0–200 cm, and 0–300 cm) over the Third Pole produced by Wang et al. (2021) was also used.

Soil erosion data came from the dataset of soil erosion intensity with 300 m resolution in Tibetan Plateau (1992, 2005, 2015) (Zhang, 2019). It was calculated using China Soil Loss Equation (CSLE) considering precipitation, soil erodibility, slope length, slope, vegetation cover, management and crop factors. Soil erodibility was also included in the dataset, and the Soil Erodibility Dataset of Pan-Third Pole in 2020 (Yang et al., 2019) was used. The soil water content data was derived from the Global High-Resolution Soil-Water Balance dataset (Trabucco and Zomer, 2010), which defines the fraction of soil water content available for evapotranspiration processes (as a percentage of the maximum soil water content) and is therefore a measure of soil water stress. We calculated catchment-level soil water content at monthly and annual scales. The mean annual ground temperature (MAGT) was derived from the MAGT and permafrost thermal stability dataset over Tibetan Plateau from 2005 to 2015 (Ran et al., 2019).

The geology characteristics include lithological class, subsurface permeability and porosity. The lithological classes came from the Global Lithological Map (GLiM) database V 1.0 (Hartmann and Moosdorf, 2012). GLiM consists of three classification levels, and the first level which contains 16 lithological classes was adopted. The subsurface permeability and porosity, two crucial parameters for groundwater modelling, were derived from the GLobal HYdrogeology MaPS 2.0 (GLHYMPS 2.0) dataset (Huscroft et al., 2018). Permeability measures how easy the rock permits the passage of fluids, and porosity measures how much water that can be stored in the subsurface. These two parameters were estimated based on the GLiM lithological map, which can differentiate fine and coarse-grained sediments and sedimentary rocks. To calculate the catchment-level characteristics, the arithmetic mean was used for porosity, while the logarithmic scale geometric mean was used for permeability.

Figure 10 shows the spatial distribution of soil & geology characteristics for inter-lake catchments on the TP. The pH value is high in the west of the Inner TP, and SOC and total nitrogen content are low in this region. The sand content is high in the south of the Inner TP and the northeast of the TP, while the clay content shows the opposite pattern. The fraction of permafrost extent is high in the north of the Inner TP. The lithological classes show a latitudinal distribution, and the main types include siliciclastic sedimentary rocks, mixed sedimentary rocks, and unconsolidated sediments. The subsurface permeability is higher in the south of the TP than in the north, and the subsurface porosity is higher in the north of the TP and the valley between the Kunlun and Gangdise Mountains in the south of the Inner TP.

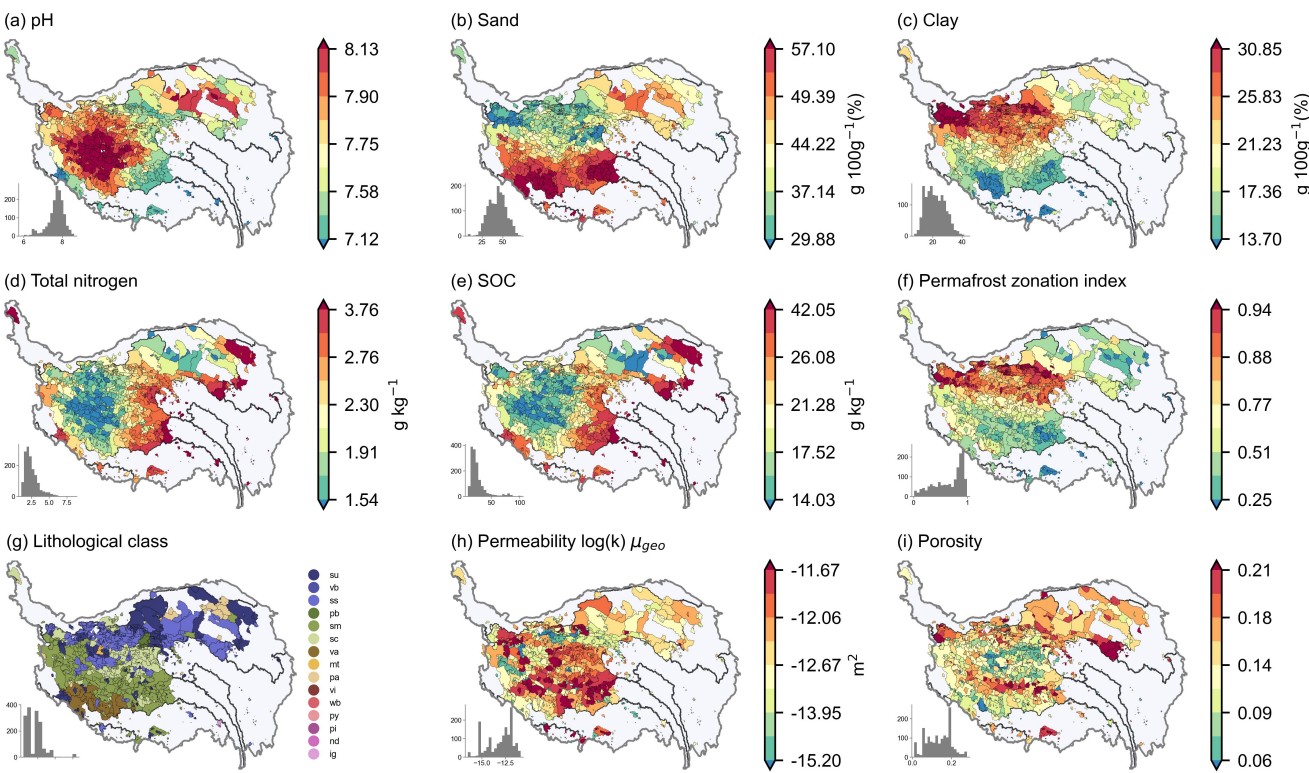

**Figure 10: Spatial distribution of soil & geology characteristics for inter-lake catchments on the TP.**

### 4.7 Anthropogenic activity characteristics

Population count and density, man-made objects such as cities and roads, and nighttime lights were used to characterize human activities in a catchment. Human footprint, a comprehensive index for evaluating human activities, was also included in this dataset. The population count and density data were obtained from the Gridded Population of the World (GPW) database v4.11 (Center for International Earth Science Information Network - Columbia University, 2018). This database provides estimates of the human population (number of persons per pixel) at a spatial resolution of 30 arc-seconds for the years 2000, 2005, 2010, 2015, and 2020. Nighttime Lights (NLI) are a useful proxy to characterise the intensity of human activity, and the DMSP-OLS Nighttime Lights v4 dataset (Doll, 2008) was used in this study. It was produced using cloud-

free remote-sensing images from the Defense Meteorological Satellite Program (DMSP) Operational Linescan System (OLS) at a spatial resolution of 30 arc-seconds. The values in this dataset represent the product of the average visible band digital number of cloud-free light detections and the percent frequency of light detection.

Road density was derived from the Global Roads Inventory Project (GRIP) dataset (Meijer et al., 2018). Nearly 60 geospatial datasets on road infrastructure (from 1997 to current) were gathered, harmonized and integrated into a global road dataset. The resulting dataset includes over 21 million km of roads, classified into five types. In this research, catchment-level road density was calculated from a simplified grid dataset at 5 arc-minute spatial resolution. Human Footprint is a measure of how much we are using the Earth's natural resources, and the Global Human Footprint v2 dataset at a spatial resolution of 30 arc-seconds (Venter et al., 2016) was used by us. In this dataset, eight different factors, including built environments, population density, electric infrastructure, crop lands, pasture lands, roads, railways, and navigable waterways, were combined to measure the direct and indirect human pressures on the environment globally in 1993 and 2009.

Figure 11 shows the spatial distribution of anthropogenic activity characteristics for inter-lake catchments on the TP. The population density, road density, and human footprint all suggest that human activities are relatively intense in the south and northeast of the TP and there is almost no human activity in the north of the Inner TP where elevation is high and environmental conditions are harsh.

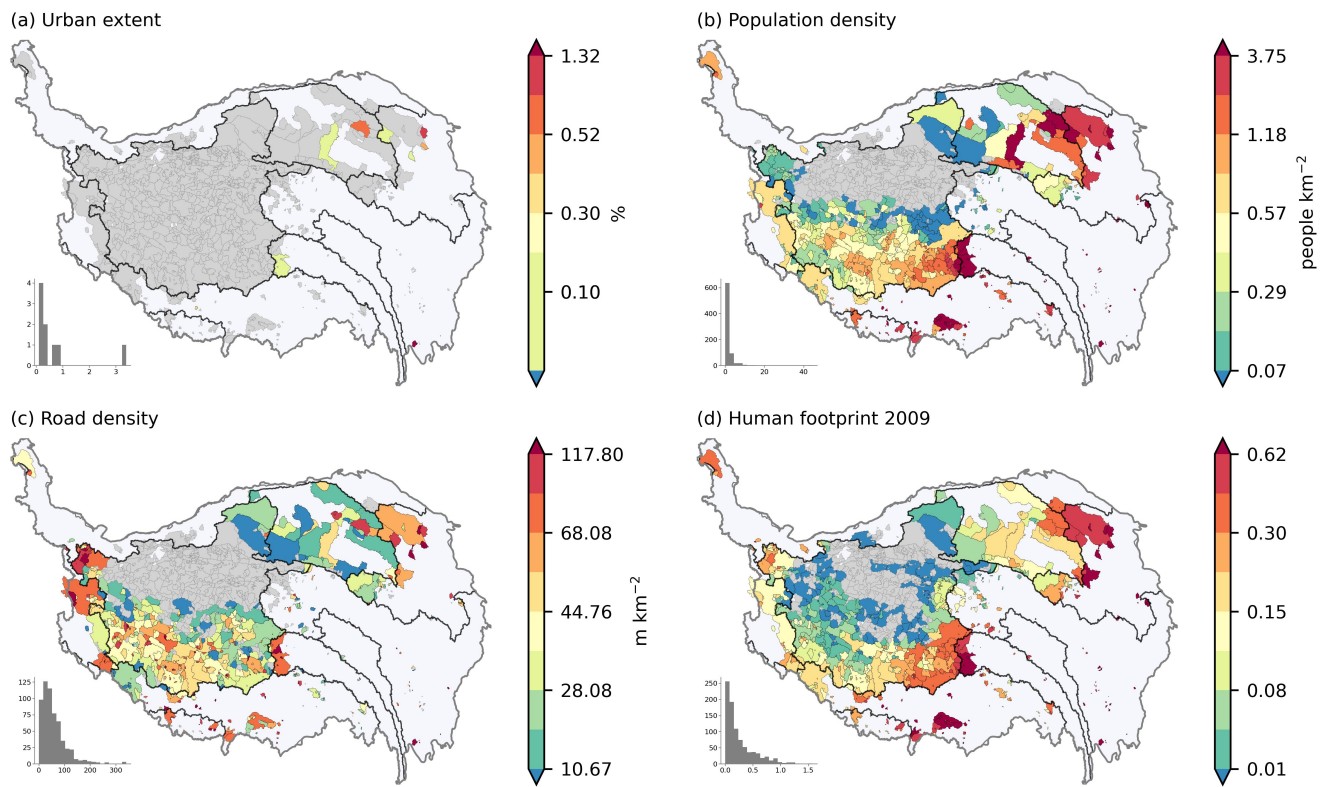

**Figure 11: Spatial distribution of anthropogenic activity characteristics for inter-lake catchments on the TP.**

## 4.8 Hydrological and meteorological time series

This dataset also provides the time series of several important hydrological and meteorological variables (Table S2), including: 1) daily meteorological variables (i.e. 2-meter air temperature, surface pressure, and specific humidity, 10-meter wind speed, downward shortwave radiation, downward longwave radiation, and precipitation amount) from the CMFD dataset covering the period 1979-2018 (Yang and He, 2019), 2) remote-sensed submonthly water level and volume data (2000-2017) extracted from Landsat images and altimetry data based on lake shoreline positions (Li et al., 2019), ~monthly water level data (2010-2020) extracted from multi-sensor altimetry data (Xu et al., 2022), lake area and mass change data at five-year intervals (1976-2020) extracted from satellite stereo and multispectral images (Zhang et al., 2021), 3) remote-sensed daily fractional snow cover based on the MODIS surface reflectance product MO/YD09GA covering the period 2000-2022 (Jiang et al., 2022), daily snow depth data (1980-2019) produced through the fusion of five gridded snow depth datasets using machine learning methods (Che et al., 2021), and daily snow water equivalent data (2002-2011) based on AMSR-E brightness temperature (Qiu, 2018b), 4) yearly glacier mass change rates (2000-2019) extracted from large-scale and openly available satellite and airborne elevation datasets (Hugonnet et al., 2021), 5) decadal maximum freezing depth data of seasonal frozen-soil (1961-2020) produced by the support vector regression model based on in-situ measurements from 2001 to 2010 and spatial environmental variables (Wang and Ran, 2021). These time series data facilitate the analysis of temporal variation at the catchment scale and can be used for hydrological modelling based on lumped hydrological models or machine learning methods.

## 5 Uncertainties of the dataset

Since the catchment-scale attributes in this dataset were mostly derived from existing datasets by calculating zonal statistics (such as sums, means, and medians), uncertainties of source datasets were propagated to the results and determined the uncertainties of this dataset. We have done our best to collect the most reliable datasets to date and will regularly update the related datasets in the future to ensure their timeliness. Still, users of this dataset need to be aware of the uncertainties of the main source datasets which are listed as follows.

1) Lake water level and volume. The RMSE (root mean square error) of the Landsat-derived water levels from Li et al. (2019) was 0.11 m. The water level data from Xu et al. (2022) had $R^2 > 0.80$ and RMSE < 0.12 m in Qinghai Lake. The uncertainties for each value in the time series of Li et al. (2019), Zhang et al. (2021), and Xu et al. (2022) can be found in corresponding uncertainty files (Table S2).

2) Topographic data. Most topographic attributes in this dataset were derived from MERIT DEM and MERIT Hydro (flow direction map) datasets. MERIT DEM was produced by eliminating main error components (e.g. absolute bias, stripe noise, speckle noise, and tree height bias) from existing DEMs (SRTM3 DEM, AW3D DEM, and VFP-DEM). It has a resolution of 3″ (~90 m at the equator) and the land areas mapped with ±2 m or better vertical accuracy were 58% (Yamazaki et al., 2017). MERIT Hydro was derived from MERIT DEM and water body datasets (G1WBM, Global

Surface Water Occurrence, and OpenStreetMap). The relative error of MERIT Hydro in drainage area delineation was less than 0.05 for 90% of Global Runoff Data Center (GRDC) gauges.

3)    Climatic data. The CMFD meteorological dataset used in this study was produced through fusion of remote sensing products, reanalysis datasets, and in-situ observations from a larger number of stations. Its accuracy in western China was validated based on independent observations, and the results showed that CMFD had closer-to-zero MBE (mean

bias error), lower RMSE, and higher $R^2$ than the Global Land Data Assimilation System (GLDAS) for almost all meteorological variables (He et al., 2020).

4)    Land cover/use data. The land cover/use data used in this study came from the fusion of six popular land use products, with an accuracy of 88.71% (Xu, 2019). The GPP and NPP data came from the MODIS products (MOD17A2H.006 and MOD17A3HGF.006). The $R^2$ between monthly MODIS GPP and eddy covariance measurements was reported to be

0.64 on average, and the RMSE was 2.55 g C m$^{-2}$ day$^{-1}$ in alpine grassland, which is the most widely-distributed biome on the TP (Zhu et al., 2018), and the $R^2$ between MODIS NPP and in-situ observations in 23 stations across China was reported to be 0.81, and the RMSE was 73.44 g C m$^{-2}$ (Sun et al., 2021). The RMSE of fractional snow cover data from Jiang et al. (2022) was 0.14 taking the results from high-resolution Landsat images as reference. The $R^2$ between snow depth data from Che et al. (2021) and in-situ observations was 0.81, and the RMSE and MAE were 7.7 cm and 2.7 cm.

5)    Soil data. The SoilGrids 2.0 dataset used in this study was generated by machine learning methods, using approximately 240 000 soil observations worldwide and over 400 environmental variables as inputs. It provides a spatial distribution map of data uncertainty generated by the quantile regression forest prediction model, which is the ratio of the interquartile range (i.e. the difference between 0.95 quantile and 0.05 quantile) over the median (Poggio et al., 2021). The catchment-level average uncertainty for each soil variable was calculated and included in this dataset. For the

maximum freezing depth of seasonal frozen-soil, the $R^2$ in the four periods of 1980s, 1990s, 2000s and 2010s were 0.77, 0.83, 0.73 and 0.71, respectively (Wang and Ran, 2021).

## 6 Data availability

The dataset of lake-catchment characteristics for the Tibetan Plateau (LCC-TP v1.0) is accessible at the National Tibetan Plateau/Third Pole Environment Data Center (https://doi.org/10.11888/Terre.tpdc.272026, Liu, 2022) and the figshare

website (https://figshare.com/articles/dataset/A_dataset_of_lake-catchment_characteristics_for_the_Tibetan_Plateau_v1_0_/20222178). There are two types of data in this dataset: spatial data stored in shapefile format and attribution data stored in csv format. The spatial data is stored in the "spatial_data" folder, including the spatial distribution of lakes (lakes.shp), the spatial extent of full catchments (full_catchments.shp), the spatial extent of inter-lake catchments (inter-lake_catchments.shp), and the flow paths among upstream and downstream lakes

(flowpath.shp). The attributes of lakes and their lake catchments are stored in LCC-TP_attributes.csv, which can be linked to the spatial data through the "LakeID" field. The time series of daily meteorological data from 1979 to 2018 are stored in the

csv files in the "time_series" folder. Each column in the csv file, except for the first one, corresponds to the data of a lake, and the column name is lake ID. The name of each file consists of two parts, connected by an underscore. The first part specifies the spatial extent which can be lake body (LK), full catchments (FC) and inter-lake catchments (IC). The second part specifies the type of meteorological variable which can be temp (temperature, K), prec (precipitation, mm), wind (wind speed, m/s), pres (air pressure, Pa), LRAD (long wave radiation, $W/m^2$), SRAD (short wave radiation, $W/m^2$) and Shum (specific humidity, kg/kg).

## 7 Conclusions

This study constructed the first dataset of lake-catchment characteristics for 1525 lakes with areas from 0.2 to 4503 $km^2$ on the TP (LCC-TP v1.0). The catchment-level characteristics were extracted for both inter-lake catchments and full catchments of lakes, and there are six categories (i.e. lake body, topographic, climatic, land cover/use, soil & geology, and anthropogenic activity) and a total of 721 attributes in the dataset. Besides multi-year average attributes, the daily time series of climatic variables were also extracted, which can be used to drive lumped hydrological models or machine learning models to simulate hydrological processes. The LCC-TP dataset contains fundamental information for analysing the impact of the catchment on lakes, which on the one hand can deepen our understanding of the drivers of lake environment change, and on the other hand, can be used to predict the water and sediment properties in unsampled lakes based on limited samples and the catchment-level attributes provided by our dataset. This provides exciting opportunities for lake studies in a spatially-explicit context and promotes the development of landscape limnology on the TP.

## Author contributions

JL and PF designed the study and wrote the manuscript. JL, PF and YQ wrote related programs and constructed the dataset. LZ, ZD, GT, PL, MJ and YL performed analysis based on the dataset. All authors contributed to the writing and editing of this paper.

## Competing interests

The authors declare that they have no conflict of interest.

## Financial support

This work was supported by the National Key Research and Development Program of China (2019YFC1509103), the National Natural Science Foundation of China (42171132 and 41930102), and the Second Tibetan Plateau Scientific Expedition and Research program (2019QZKK0503).

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
