# Peer review of "A dataset of lake-catchment characteristics for the Tibetan Plateau"

_Earth System Science Data, 2022_

## Author Comment (AC3)

This dataset of Tibetan Plateau lake-catchment characteristics fills a data gap for conducting a variety of potential scientific studies on the lakes in the region, which itself is of great importance to understanding the earth systems. Development of the dataset is well justified and well designed. The dataset is comprehensive and contains many aspects of information on a large number of lakes over a long period of time (including time series data). Besides, potential users of the dataset would appreciate the distinction between full-lake-catchment and inter-lake-catchment statistics.

The only suggestion from the reviewer is to add a section to briefly discuss or highlight the uncertainties of such a dataset. For example, given that most of the environmental statistics were obtained from existing datasets (DEM, Land cover, climatic data, etc.), highlighting sources/magnitude of uncertainties of such datasets, and discussing how the uncertainties propagate to the developed TP lake-catchment dataset would be beneficial to future users of this dataset.

Overall, the reviewer thinks the dataset is scientifically sound (adding discussion on uncertainty is a plus) and presented in a very clear manner. This dataset could be of great interest to the earth scientfic reserach communities.

Reply:
We appreciate the reviewers' insightful and helpful comments on our manuscript. We have revised the manuscript according to the reviewer's suggestion. A new section about the uncertainties of the dataset has been added in the revised manuscript (L330-366) to facilitate the usage of this dataset.

**"5 Uncertainties of the dataset**
Since the catchment-scale attributes in this dataset were mostly derived from existing datasets by calculating zonal statistics (such as sums, means, and medians), uncertainties of source datasets were propagated to the results and determined the uncertainties of this dataset. We have done our best to collect the most reliable datasets to date and will regularly update the related datasets in the future to ensure their timeliness. Still, users of this dataset need to be aware of the uncertainties of the main source datasets which are listed as follows.

1) Lake water level and volume. The RMSE (root mean square error) of the Landsat-derived water levels from Li et al. (2019) was 0.11 m. The water level data from Xu et al. (2022) had $R^2 > 0.80$ and RMSE < 0.12 m in Qinghai Lake. The uncertainties for each value in the time series of Li et al. (2019), Zhang et al. (2021), and Xu et al. (2022) can be found in corresponding uncertainty files (Table S2).

2) Topographic data. Most topographic attributes in this dataset were derived from MERIT DEM and MERIT Hydro (flow direction map) datasets. MERIT DEM was produced by eliminating main error components (e.g. absolute bias, stripe noise, speckle noise, and tree height bias) from existing DEMs (SRTM3 DEM, AW3D DEM, and VFP-DEM). It has a resolution of 3″ (~90 m at the equator) and the land areas mapped with ±2 m or better vertical accuracy were 58% (Yamazaki et al., 2017). MERIT Hydro was derived from MERIT DEM and water body

datasets (G1WBM, Global Surface Water Occurrence, and OpenStreetMap). The relative error of MERIT Hydro in drainage area delineation was less than 0.05 for 90% of Global Runoff Data Center (GRDC) gauges.

3) Climatic data. The CMFD meteorological dataset used in this study was produced through fusion of remote sensing products, reanalysis datasets, and in-situ observations from a larger number of stations. Its accuracy in western China was validated based on independent observations, and the results showed that CMFD had closer-to-zero MBE (mean bias error), lower RMSE, and higher $R^2$ than the Global Land Data Assimilation System (GLDAS) for almost all meteorological variables (He et al., 2020).

4) Land cover/use data. The land cover/use data used in this study came from the fusion of six popular land use products, with an accuracy of 88.71% (Xu, 2019). The GPP and NPP data came from the MODIS products (MOD17A2H.006 and MOD17A3HGF.006). The $R^2$ between monthly MODIS GPP and eddy covariance measurements was reported to be 0.64 on average, and the RMSE was 2.55 g C $m^{-2}$ $day^{-1}$ in alpine grassland, which is the most widely-distributed biome on the TP (Zhu et al., 2018), and the $R^2$ between MODIS NPP and in-situ observations in 23 stations across China was reported to be 0.81, and the RMSE was 73.44 g C $m^{-2}$ (Sun et al., 2021). The RMSE of fractional snow cover data from Jiang et al. (2022) was 0.14 taking the results from high-resolution Landsat images as reference. The $R^2$ between snow depth data from Che et al. (2021) and in-situ observations was 0.81, and the RMSE and MAE were 7.7 cm and 2.7 cm.

5) Soil data. The SoilGrids 2.0 dataset used in this study was generated by machine learning methods, using approximately 240 000 soil observations worldwide and over 400 environmental variables as inputs. It provides a spatial distribution map of data uncertainty generated by the quantile regression forest prediction model, which is the ratio of the interquartile range (i.e. the difference between 0.95 quantile and 0.05 quantile) over the median (Poggio et al., 2021). The catchment-level average uncertainty for each soil variable was calculated and included in this dataset. For the maximum freezing depth of seasonal frozen-soil, the $R^2$ in the four periods of 1980s, 1990s, 2000s and 2010s were 0.77, 0.83, 0.73 and 0.71, respectively (Wang and Ran, 2021)."

---

## Author Comment (AC4)

The authors provide the Tibetan Plateau lake-catchment characteristics dataset, which contains a wide range of information such as topographic, climatic, land characteristics, and anthropogenic activity characteristics. The dataset will be a valuable input for relevant hydrologic or climatic studies in the region. The manuscript is generally well written, clear, and easy to follow. I have only a few comments on the manuscript.

Reply:
We appreciate the reviewers' positive and helpful comments on our manuscript. We have addressed all of the concerns of the reviewer in the revised manuscript, and the point-by-point responses are given below.

General comments:

The authors extract catchment attributes from various exiting datasets, such as MERIT Hydro dataset, SoilGrids, CMFD. Please add more descriptions of how those data are generated/obtained (are they observation-based? model-processed? machine learning-based?).

Reply: We have added more descriptions of how the used datasets were generated/obtained (L341-344, L347-348, L360-361, and L317-327).

L341-344: "MERIT DEM was produced by eliminating major error components (e.g. absolute bias, stripe noise, speckle noise, and tree height bias) from existing DEMs (SRTM3 DEM, AW3D DEM, and VFP-DEM). It has a resolution of 3″ (~90 m at the equator) and the land areas mapped with ±2 m or better vertical accuracy were 58% (Yamazaki et al., 2017)."

L347-348: "The CMFD meteorological dataset used in this study was produced through fusion of remote sensing products, reanalysis dataset and in-situ observations."

L360-361: "The SoilGrids 2.0 dataset used in this study was generated by machine learning methods, utilizing approximately 240 000 soil observations worldwide and more than 400 environmental variables as inputs."

L317-327: "2) remote-sensed submonthly water level and volume data (2000-2017) extracted from Landsat images and altimetry data based on lake shoreline positions (Li et al., 2019), ~monthly water level data (2010-2020) extracted from multi-sensor altimetry data (Xu et al., 2022), lake area and mass change data at five-year intervals (1976-2020) extracted from satellite stereo and multispectral images (Zhang et al., 2021), 3) remote-sensed daily fractional snow cover based on the MODIS surface reflectance product MO/YD09GA covering the period 2000-2022 (Jiang et al., 2022), daily snow depth data (1980-2019) produced through the fusion of five gridded snow depth datasets using machine learning methods (Che et al., 2021), and daily snow water equivalent data (2002-2011) based on AMSR-E brightness temperature (Qiu et al., 2018b), 4) yearly glacier mass balance data (2000-2019) leveraging large-scale and openly available satellite and airborne elevation datasets (Hugonnet et al., 2021), 5) the decadal maximum freezing depth data of seasonal frozen-soil (1961-2020) produced by the support vector regression model based on in-situ measurements from 2001 to 2010 and spatial environmental variables (Wang and Ran, 2021)."

Given the use of existing datasets for obtaining the lake characteristics, I think the first part of the

data development, i.e. catchment delineation is the critical step in your data development. Can you explain more about what kind of methods could be applied, and why you chose your method (instead of using Liu et al, 2020, which already exists) and any limitations/uncertainty of the method?

Reply: Traditional river-oriented catchment delineation methods are not suitable for the delineation of lake catchments. Liu et al (2020) proposed a lake-oriented approach to delineating endorheic catchments, which can be used to delineate the full catchments of endorheic lakes in this study. But there are more tasks in this study, including the delineation of both full catchments and inter-lake catchments for endorheic lakes and upstream lakes, the construction of topological relationship among lakes/lake-catchments, and the tracing of flow path among upstream and stream lakes. Therefore, we developed a software using the C and Python programming language to implement these tasks, and the source code are open (https://github.com/LoserOne-ovo/basin_delineation). This has been explained in the revised manuscript (L101-107).

I am not sure if I did something wrong, but I failed to connect to FTP to see the LCC-TP. I got the error saying "Home directory not available - aborting". - can you provide the full path of the dataset?

Reply: This dataset can be visited at the following FTP, server: 210.72.14.198, username: download_366550, password: 15147322. We also uploaded the dataset to the figshare website: https://figshare.com/articles/dataset/A_dataset_of_lake-catchment_characteristics_for_the_Tibetan _Plateau_v1_0_/20222178.

Specific comments:

Fig.2: it would be helpful if you mark the first, second, and third steps (L82; "Three steps") of the procedure also in the figure.

Reply: The first, second, and third steps have been marked in Fig. 4.

[Figure]

Fig.4: until when does the "continue" repeat (i.e. from "stack is empty?" to "push its upstream pixels into tack")?

Reply: Thank you very much for carefully reading the chart and providing the nice suggestion. We have revised the flowchart to make it clearer. There are two nested loops in the flowchart: for the external loop, if all the pixels are visited, the loop will stop; for the inner loop, if the stack is empty, the loop will stop.

[Figure]

L165-167: do you mean you provide multiple relief values computed with different window sizes? What is the purpose of using multiple window sizes?

Reply: Yes, we computed multiple relief values using different window sizes (i.e. 5×5, 11×11, 21×21, 31×31, 41×41, and 51×51) in this study. For the researches focusing on small-scale terrain variation, a small window size is appropriate; when the focus is large-scale terrain variation, a larger window size is preferred. To meet the needs of different analysis scenarios, we provide multiple relief values computed with different window sizes. This has been explained in the revised manuscript (L180-182).

---

## Author Comment (AC5)

Lakes on the Tibetan Plateau are important for water cycle studies. There are many previous studies understanding the lake area variation and their influencing factors. But there are still lack of systematic studies including the lake-catchment characteristics. In this study, the authors put many attributes together, which has great potential to help us understand the characteristics of lake-catchments on the TP in a systematic way. I find this study is novel, and has potential to be an important paper for process understanding on the TP lakes. But before considering for acceptance I have some concerns.

Reply: We appreciate the reviewers' positive and helpful comments on our manuscript. We have addressed all of the concerns of the reviewer in the revised manuscript, and the point-by-point responses are given below.

1. Although the streamflow observations on the TP is very limited, I suggest the authors to include more time-series of the hydrologic and meteorological related attributes. Hydrologic related data may include the lake area and volume temporal variation, the area and volume changes of glaciers, the changes of the proportion of permafrost and seasonal frozen-soil. Meteorological data can be even reanalysis data, e.g. daily precipitation and air temperature. This will make this study more systematic. These new chronical attributes will allow us to do more process understanding on the lake area changes.

Reply: According to the reviewer's suggestion, we have added more time-series data to the dataset. The revised dataset includes time series of 16 hydrological and meteorological variables, which are listed as follows:

1) daily meteorological variables (i.e. 2-meter air temperature, surface pressure, and specific humidity, 10-meter wind speed, downward shortwave radiation, downward longwave radiation, and precipitation amount) from the CMFD dataset covering the period 1979-2018 (Yang and He, 2019);

2) remote-sensed submonthly water level and volume data (2000-2017) extracted from Landsat images and altimetry data based on lake shoreline positions (Li et al., 2019), ~monthly water level data (2010-2020) extracted from multi-sensor altimetry data (Xu et al., 2022), lake area and mass change data at five-year intervals (1976-2020) extracted from satellite stereo and multispectral images (Zhang et al., 2021);

3) remote-sensed daily fractional snow cover based on the MODIS surface reflectance product MO/YD09GA covering the period 2000-2022 (Jiang et al., 2022), daily snow depth data (1980-2019) produced through the fusion of five gridded snow depth datasets using machine learning methods (Che et al., 2021), and daily snow water equivalent data (2002-2011) based on AMSR-E brightness temperature (Qiu, 2018b);

4) yearly glacier mass change rates (2000-2019) extracted from large-scale and openly available satellite and airborne elevation datasets (Hugonnet et al., 2021);

5) decadal maximum freezing depth data of seasonal frozen-soil (1961-2020) produced by the support vector regression model based on in-situ measurements from 2001 to 2010 and spatial environmental variables (Wang and Ran, 2021).

The published dataset has been updated to include these time series data (https://doi.org/10.11888/Terre.tpdc.272026 and https://figshare.com/articles/dataset/A_dataset_of_lake-catchment_characteristics_for_the_Tibetan

_Plateau_v1_0_/20222178), and a new section "Section 4.8 Hydrological and meteorological time series" and "Table S2. The descriptions about the time series data in the LCC-TP dataset (version 1.0)" have been added to the revised manuscript to describe the related datasets (L313-329).

2. What does precipitation rate mean? What does "fractional snow cover" mean? Snow is an important component in cold region hydrology. Is it possible to give the time-series of fractional snow cover as another attribute? Is there snow water equivalent data available?

Reply: Precipitation rate refers to the amount of precipitation per unit time. The unit in the source data (He et al., 2020) is mm hr$^{-1}$. The daily, monthly and annual precipitation data in this dataset were aggregated from hourly data, and the unit of aggregated precipitation amount is mm. For clarity, "precipitation rate" has been rephrased to "precipitation amount" in the revised manuscript (L141, 193, and 316).

The phrase "fractional snow cover" means the fraction of a pixel that is snow-covered, which has been explained in the revised manuscript (L233-234).

According to the reviewer's suggestion, the time series data of fractional snow cover, snow depth and snow water equivalent have been added (L320-324 and L357-359).